# Conflicts of Interest and Misleading Statements in Official Reports about the Health Consequences of Radiofrequency Radiation and Some New Measurements of Exposure Levels

**Susan Pockett**

School of Psychology, University of Auckland, Auckland 1142, New Zealand; s.pockett@auckland.ac.nz

**Abstract:** Official reports to governments throughout the Western world attempt to allay public concern about the increasing inescapability of the microwaves (also known as radiofrequency radiation or RF) emitted by "smart" technologies, by repeating the dogma that the only proven biological effect of RF is acute tissue heating, and assuring us that the levels of radiation to which the public are exposed are significantly less than those needed to cause acute tissue heating. The present paper first shows the origin of this "thermal-only" dogma in the military paranoia of the 1950s. It then reveals how financial conflict of interest and intentionally misleading statements have been powerful factors in preserving that dogma in the face of now overwhelming evidence that it is false, using one 2018 report to ministers of the New Zealand government as an example. Lastly, some new pilot measurements of ambient RF power densities in Auckland city are reported and compared with levels reported in other cities, various international exposure limits, and levels shown scientifically to cause biological harm. It is concluded that politicians in the Western world should stop accepting soothing reports from individuals with blatant conflicts of interest and start taking the health and safety of their communities seriously.

**Keywords:** radiofrequency radiation; RF; microwave; cellphone; smart technology; public health; cancer; diabetes; depression; dementia

## 1. Introduction

The health effects of radiofrequency radiation (RF) emitted by 'smart' technologies have become a topic of significant public concern throughout the world. Official reports prepared for governments in the English-speaking world tend to be aimed at allaying what the report writers paint as unjustified fears, by assuring the public and their elected representatives that scientific research in this area shows no reason for concern about emissions that abide by current regulatory guidelines.

The present paper has three goals: (1) to document the prevalence of undisclosed conflicts of interest, both in the original setting of regulatory guidelines and among the authors of one representative government report [1] defending these; (2) to assess the accuracy of certain key statements in that report; and (3) to provide the first publicly available raw power density readings from two specific sites in Auckland City, as part of a pilot study on how Auckland readings compare with (i) readings in other Western cities, (ii) the recommended limit on public exposure in New Zealand, (iii) the recommended limits on public exposure in various other countries, and (iv) the power densities of RF shown in the scientific literature to have harmful biological effects.

## 2. Conflict of Interest: A History

The 2018 version of the guidelines document put out by the International Committee of Medical Journal Editors [2] defines conflict of interest as follows:

"A conflict of interest exists when professional judgment concerning a primary interest (such as patients' welfare or the validity of research) may be influenced by a secondary interest (such as financial gain). . . . Financial relationships (such as employment, consultancies, stock ownership or options, honoraria, patents and paid expert testimony) are the most easily identifiable conflicts of interest and the most likely to undermine the credibility of the journal, the authors and science itself . . . . Purposeful failure to disclose conflicts of interest is a form of misconduct."

By this definition, conflict of interest has been a constant feature at all levels of discourse in the area of RF exposure standards, from the original setting of the "Guidelines" now used by regulators all over the Western world, right down to the authorship of individual government reports in the present day.

History in this area begins nearly 70 years ago. Currently-entrenched official positions on safe RF public exposure levels originated in the 1950s, during the period known as the Cold War between the United States of America (USA) and the Union of Soviet Socialist Republics (USSR). At this time, the US Department of Defense (DOD) was charged with developing radar (radio-based detection and ranging) systems capable of detecting incoming Soviet missiles. This meant that the US military had a major vested interest in producing radar installations that were as powerful as possible. Objections raised by local US communities upset at the unheralded appearance of such facilities in their neighborhoods were dismissed as a minor cost in comparison with the perceived benefit of preventing nuclear annihilation. Thus, in terms of the above definition of conflict of interest, both of the interests that were clearly in conflict here were perceived to be protection of the public—it is just that one of them (the one that prevailed) was a product of military paranoia, while the other involved the much more mundane question of everyday health and safety.

A further complication during this historical period was that microwaves were widely used in diathermy, a then popular medical treatment for a number of conditions thought to be improved by tissue heating. Hence, it was convenient for both military and medical circles in the US to ignore early scientific indications to the contrary and choose to believe uncritically the hypothesis that the only way in which microwave radiation could affect biological organisms was by heating them. Interestingly however, when it came to the setting of standards regulating the level of microwave radiation to which people could safely be exposed, the medical profession was deemed to have too much vested interest in diathermy to participate, while the obvious conflict of interest involved in making the military responsible for setting acceptable microwave power limits was ignored [3]. By 1960, all three branches of the US military had concluded, on the basis of one man's calculations and some minimal experimentation (involving disruption of food-motivated behavior in irradiated laboratory animals) that 10 mW/cm$^2$ was a safe power density limit to prevent excessive tissue heating, and after some debate, this figure duly became the basis of the first IEEE/ANSI C95.1 microwave standard in 1966. Thereafter, the DOD treated all reports of biological effects of RF power densities less than 10 mW/cm$^2$ as a threat to national security and shut down any lab that produced them [4–6].

In contrast, the Soviets whose imagined missiles the DOD was charged with detecting and destroying concentrated on following up early reports of sub-thermal microwave effects, and as a result, set their exposure limit at 0.01 mW/cm$^2$. This thousand-fold stricter limit posed a serious problem for US military planners—if any of America's western European allies were tempted to adopt it, deployment of American radar installations in Europe would be jeopardized. Therefore, concurrent with the space/arms race, an RF standards race was played out in various international organizations, such as WHO (the World Health Organization) and NATO (the North Atlantic Treaty Organization) [3]. Internationalization of what was by now the unchallengable dogma that tissue

heating was the only possible biological effect of RF was achieved by the simple expedient of embedding individuals committed to the thermal-only narrative in WHO and NATO. In 1971, Sol Michaelson, the American who had been most instrumental in the adoption of the thermal-only standard by ANSI C95.1, was appointed to a committee called the Task Group on Environmental Health Criteria for Radiofrequency and Microwaves, jointly convened by WHO and the International Radiation Protection Agency (IRPA). The founding chairman of IRPA was Michael Repacholi, an Australian also committed to the thermal-only dogma. In 1992, IRPA morphed into ICNIRP (the International Commission on Non Ionizing Radiation Protection), with Repacholi still as the chair. And in 1998, ICNIRP brought out the Guidelines document which still enshrines the ANSI thermal-only dogma as the basis of national standards throughout the English-speaking world.

Meanwhile, back in the USA, a second strand of activity in support of the thermal-only dogma was quietly emerging. In the early 1970s, a growing popular environmental movement and the consequent espousal by the US Environmental Protection Agency (EPA) of a precautionary approach to a great many potential health hazards were seen by corporate interests as a threat to the foundations of industrial society [7]. The challenge for industry was cast as how best to respond to legislative restrictions on the activities of corporations—and in particular to the science that led to those restrictions. One major response to this challenge was the establishment in 1972 of a 'Business Roundtable' consisting of many of America's CEOs, for the express purpose of promoting "less unwarranted intrusion by government into business affairs" and ensuring that "the business sector in a pluralistic society should play an active and effective role in the formation of public policy" [7].

Lobby offices were established in Washington, and a number of industry-backed think tanks created to come up with strategies applicable to all industries. Measures adopted with respect to the biological effects of microwave emissions mirrored those of the tobacco industry. They included the following:

- *Creation of an air of uncertainty about the science*: Given that biological organisms are formidably complex and that science by its nature rarely involves complete certainty, this should perhaps not have proved too difficult. But just to make sure, a concerted campaign of disinformation was launched anyway. Basically, whenever a piece of science inimical to industry or Air Force interests appeared, contractors were hired to discredit it by *apparently* repeating the experiments, but actually changing critical factors to produce more funder-friendly results. Frey [6] describes one such attempt as follows:

  "After my colleagues and I published in 1975 [8], that exposure to very weak microwave radiation opens the regulatory interface known as the blood brain barrier (BBB), a critical protection for the brain, the Brooks AFB group selected a contractor to supposedly replicate our experiment. For 2 years, this contractor presented data at scientific conferences stating that microwave radiation had no effect on the BBB. After much pressure from the scientific community, he finally revealed that he had not, in fact, replicated our work. We had injected dye into the femoral vein of lab rats after exposure to microwaves and observed the dye in the brain within 5 min. The Brooks contractor had stuck a needle into the animals' bellies and sprayed the dye onto their intestines. Thus it is no surprise that when he looked at the brain 5 min later, he did not see any dye; the dye had yet to make it into the circulatory system."

The continuing nature of such campaigns is suggested by Maisch [3], who writes:

  "A survey conducted by the New York based publication Microwave News in 2006 consisted of examining papers on microwave effects on DNA that were published in peer-reviewed journals since 1990. A total of 85 papers on the topic were identified. 43 of the papers reported finding a biological effect and 42 did not. Of the 42 no-effect papers, 32 were identified as having been funded by either the U.S. Air Force or industry.

With the 43 papers that reported effects, only 3 were identified as being funded by Air Force or industry. This survey thus suggests that the source of funding has a strong influence on the outcome of research".

- *Adoption of an algebraic model of evidence assessment*: Once approximately equal numbers of papers had been installed in the scientific literature concluding that sub-thermal levels of microwaves on the one hand do, but on the other hand do not, have harmful biological effects, the narrative was promulgated in official circles that "weight of evidence" is the important thing to consider in such matters. The implicit model behind this narrative involves an unstated presumption that each negative study (i.e., each study that does not find any effect of low intensity microwaves) cancels out one positive study (i.e., one study that does find an effect of low intensity microwaves); with an algebraic sum of zero indicating no effect [9]. Any inconvenient remainder is then dealt with by impugning the validity and/or the significance of particularly convincing postive studies: as, for example, in Section 4.2 and Appendix A of the NZ Government Interagency Report 2018 [1].

- *Population of regulatory bodies by industry insiders*: The above strategies certainly served to convince time-strapped politicians that all is fine, but to an unbiased scientist, they appear decidedly dicey. Thus, the most vital of all the strategies implemented by Big Wireless has been the appointment to regulatory roles of people who are, or used to be, members of the industries they are now charged with regulating. Arguably the most important regulatory body in the world is ICNIRP, whose 1998 Guidelines document is still the basis of the national standards adopted by the governments of most English-speaking nations. ICNIRP is a self-selected, private (non-governmental) organization, populated exclusively by members invited by existing members. The organization is very concerned to project the image that it is composed of disinterested scientists—indeed all ICNIRP members are required to post on the organization's website detailed declarations of interest (DOIs). However, a closer inspection of these DOIs reveals that a good many of the sections of a good many of the forms remain unfilled, and a detailed list of undeclared conflicts of interest among ICNIRP members has been published by a group of concerned citizens [10]. The relevant section of WHO is essentially identical to ICNIRP [11]: Michael Repacholi, the founder of ICNIRP, established the WHO International EMF Project (IEMFP) in 1996 and remained in charge of it until 2006 [3], when he reportedly resigned after allegations of corruption [12] to officially become an industry consultant [13]. In 2004, Repacholi stated in a conference presentation that the IEMFP was able to "receive funding from any source through Royal Adelaide Hospital; an agency established through WHO Legal Department agreement to collect funds for the project"—an arrangement that reportedly enabled receipt of annual payments of $150,000 from the cellphone industry [3,14]. Thus, in spite of their stated rules and protestations to the contrary, there have been persistent allegations that both ICNIRP and the relevant section of WHO are riddled with undeclared conflicts of interest. In the USA, the Federal Communications Commission, whose function it is to regulate the wireless industry in that country, has been openly characterized by the Edmond J. Safra Center for Ethics at Harvard University as "a captured agency" [15].

On a much smaller scale, the New Zealand government's Interagency Committee's 2018 Report to Ministers, which is discussed in the next section of the present paper, does not specify the identities of its authors. In early 2019, a request under New Zealand's Official Information Act for the Ministry of Health to supply these names produced only a statement from someone styled "Deputy Director-General Population Health and Prevention" that "The Ministry does not usually release names as these often change, and the members represent their organisation (unlike most committees where the person is there for their specific expertise)." Fortunately however, an earlier OIA request for meeting minutes had (eventually) been more successful, yielding notes for the minutes of the 9 August 2018 meeting of the InterAgency Committee—the last meeting before the Committee's Report was released.

These notes are recorded as having been taken by the committee's acting secretary, Martin Gledhill. As well as being MOH's representative on the Committee, Martin Gledhill derives a significant portion of his income by providing RF measurement services to all the Telcos operating in New Zealand,

through an independent consultancy called EMF Services. Email correspondence between the author and Mr. Gledhill failed to reveal the precise methodology by which these measurements are made, on the grounds that the report in which this is presumably detailed is owned by SPARK (a major Telco in New Zealand), and although Mr. Gledhill asked SPARK if he could send it to the present author, they refused to release it. The EMF Services website describes Martin Gledhill as New Zealand's representative to the WHO EMF Project—the same WHO project started by Michael Repacholi, as detailed above—and a member of the IEEE International Committee on Electromagnetic Safety—the same committee that enshrined the first thermal-only standard in 1966. Thus, at least this core member of the NZ InterAgency Committee has a massive vested interest in retaining thermal-only regulatory limits.

The other members of the NZ InterAgency Committee recorded as being present at the August 9 meeting included three university scientists, four overt representatives of the wireless and telecommunications industries, and six bureaucrats representing various government departments. Of the scientists, one was an epidemiologist from the University of Otago, who has never published on the health effects of RF and according to the notes contributed only by repeatedly assuring the Committee that in his opinion, the benefits of technology outweigh the risks. The second scientist was an epidemiologist from Massey University, one of whose many research projects involves participation in the multinational MOBI-kids project; she is noted as reporting to the committee that design problems make the results of this project inconclusive. The third scientist was another epidemiologist from Massey University, one of whose many research projects involves participation in the multinational INTEROCC project, an offshoot of the controversial INTERPHONE project. He reports to the committee that INTEROCC has not found any effect of occupational RF exposure on meningiomas and is winding down; according to the meeting notes, he fails to make any mention of the methodological controversy generated by this negative finding [16–19]. No laboratory scientist—no physiologist, neuroscientist, biochemist or biophysicist, whose professional expertise might have enabled them to discuss the many scientific publications now available on the specific mechanisms by which RF affects biological organisms—was present.

No member of the committee makes any declaration about the existence or absence of individual conflicts of interest. No mention is made anywhere of the fact that the current New Zealand government, having campaigned against the TransPacific Partnership Agreement (TPPA) before the last election and then signed a renamed version of it (CPTPPA) as soon as they got into power, is seriously constrained by a realistic fear of being sued under the CPTPPA for passing any law that impacts the profits of any of the multinational corporations that promote such agreements, with the suit being settled under the investor state dispute mechanisms of the CPTPPA by a three-person international court consisting of two judges nominated by the organization that brings the suit and one by the New Zealand government.

## 3. Misleading Statements in the New Zealand Government's Interagency Committee on the Health Effects of Non-Ionizing Fields Report to Ministers 2018

This report to Ministers of the New Zealand Government could serve as a textbook example of ICNIRP spin. Almost the entire reference list consists of papers written by ICNIRP members—none of the papers cited in Section 3.2 below is cited. The report's conclusion—that the 1998 ICNIRP Guidelines document on which the current New Zealand guidelines are based is still the gold standard in the field, its thermal-only recommended exposure limit providing adequate protection for the public—gives every indication of having been predetermined. And in support of this conclusion, the report makes a number of seriously misleading statements.

Four of these statements are discussed below.

### 3.1. Misleading Statement One (p. 2)

"Animal studies do not suggest an effect of RF fields on cancer."

The wording of this statement ("effect *on* cancer") is somewhat ambiguous, but the clear intent is to convey the idea that animal studies do not suggest that RF fields can cause cancer.

The only evidence cited in support of this statement is a relatively long section devoted to acknowledging the existence, but attempting to minimize the significance, of a recent study by the National Toxicology Program (NTP) of the US Department of Health which clearly demonstrates that RF fields *do* cause cancer. According to the 19-member peer review panel that examined this study [20], its results provide "clear evidence"—the highest standard of proof—that RF fields cause schwannomas (malignant tumors of the Schwann cells that sheath all myelinated nerves) in the hearts of male rats. The NTP study also reports less clear evidence that RF causes various other tumors (gliomas in the brain, pheochromocytomas in the adrenal gland, and tumors of the prostate and pancreas). The relevant section of the NZ Interagency Report notes these facts, but concludes by citing a non-peer-reviewed ICNIRP note criticizing the methodology and minimizing the significance of the NTP study [21]. The NZ Report fails to mention a published rebuttal of the ICNIRP criticisms [22], which was accepted by the journal *Environmental Research* on 7 September 2018 (the precise stated cut-off date for publications cited by the NZ Report). The NZ Interagency Report ignores altogether a second major rodent study (available online 18 March 2018), done in a different country (Italy) by different investigators (the Ramanizzi Institute), involving 2248 rats and confirming the results of the NTP study [23].

Also mentioned but dismissed as unpersuasive is an earlier mouse study showing a facilitatory effect of lifelong exposure to RF on the development of lung, liver, kidney, and blood cancers caused by *in utero* administration of the chemical carcinogen ethyl nitrosourea [24]. The authors of that study specifically comment on the fact that this result is not dose-related with respect to RF; which actually accords well with the unexpected finding of a counterintuitive, inverted-U-shaped dose–response curve in relation to RF damage of the blood–brain barrier reported much earlier [25]. However, none of the scientists involved comments on this correspondence with earlier work: instead, the absence of the 'expected' dose–response relationship is taken as a reason for dismissing the facilitation study, by a research group who also make statements like "exposed groups were compared only to the sham-exposed control group . . . in a carcinogenesis study, it is essential to compare results to the negative control group and to in-house historical data and/or to published database(s) in the case of no or insufficient internal data." [26]. This is pure nonsense. When a scientific study finds significant differences between an exposed group and a sham-exposed group, it is disingenuous to claim that those differences are meaningless because there was no group of animals that was completely unexposed. Sham controls are universally acknowledged to be more rigorous than negative controls. However, since no laboratory scientist who could have pointed this out sits on the NZ InterAgency Committee, their report is able to use the untenable claims made in [26] to dismiss the legitimate findings reported in [24].

The least that can be said about all this is that the existence of two major rodent studies, both of which report clear evidence that RF directly causes malignant cardiac tumors, renders incorrect and misleading the statement "animal studies do not suggest an effect of RF fields on cancer". Indeed, given that a relative lack of animal evidence for carcinogenicity was the main stated reason for the IARC/WHO classification of RF as only a Group 2B ("possible") carcinogen in 2011 [27], the combination of the NTP and Ramanizzi studies must be seen as lending strong support to recent calls [28,29] for the upgrading of the IARC/WHO classification to Grade 1: "carcinogenic to humans".

*3.2. Misleading Statement Two (p. 2)*

"RF research is continuing in a number of areas, but data currently available provides no clear and persuasive evidence of any other effects."

This extraordinary statement hangs, in notably legalistic fashion, on the words "clear and persuasive evidence". Given that there are now over 2,000 peer-reviewed papers in the scientific literature documenting multiple "other effects" of RF, the obvious question is "persuasive to whom"?

The data documenting these multiple other effects clearly *were* found persuasive by the peer reviewers of the reputable scientific journals in which they are published. If the authors of this report did not find any of this evidence persuasive, one might reasonably ask "why not?"

In the absence of any alternative explanation, it seems likely that the answer to this question is simply, "because ICNIRP (and/or WHO and/or the wireless industry employers of many of the committee members) said so". Since all three of these entities have been shown to be massively invested in finding "unpersuasive" any and all reports that sub-thermal levels of RF have any biological effects at all, this answer can hardly be taken as a valid reason for ignoring and/or dismissing such a large volume of evidence; some of which is documented and briefly discussed below.

Demonstrated "other effects" of RF include:

- *Psychiatric problems, including depression*: For a review of a large number of peer-reviewed studies in this area, see [30]. Because inexplicable mental health issues among the young are an increasing problem in New Zealand, this must be seen as a rather important "other effect" of RF radiation.

- *Diabetes*: Wi-Fi irradiation of young rats causes damage to the pancreas and reduced insulin secretion [31,32] and is thus the standard method of producing an animal model of diabetes. Epidemiological evidence [33] shows statistically significant increases in pre-diabetic blood markers in human children attending a school near a cell tower, as compared with an otherwise identical group of children whose school is further from a cell tower. These findings suggest that (a) cell towers should not be built near schools and (b) Wi-Fi in schools should be replaced with cabled internet connections, accessed by multiple jack points for convenience.

- *Breakdown of the blood–brain barrier (BBB)*: Double-blind studies done as long ago as 1975 showed that RF causes abnormal leakage of fluorescein dye from the blood of rats into their brain tissue [8], and disingenuous attempts to discredit that finding constituted the first documented dirty tricks campaign in the area [6]. Honest attempts to replicate the 1975 experiments proved hard to interpret, until it was realized that a counterintuitive, inverted-U-shaped dose–response curve held—at which point it became clear that the parameters involved in mobile phone use are particularly effective in disrupting the BBB [25]. Because disruption of the BBB is a known contributor to the onset and development of Alzheimer's disease and other forms of dementia [34,35], at least two public health conclusions might reasonably be drawn from these findings. First, it would be prudent to advise the increasing population of elderly citizens to avoid cell phones, smart meters, and Wi-Fi. But perhaps more importantly, chronic exposure of the young to RF now starts in the womb and continues throughout babyhood (wireless baby monitors), childhood (wrist-worn child locators), and adolescence (smart phones, Wi-Fi). Because the biological effects of RF are known to be cumulative, urgent steps should be taken to reduce the exposure of babies, children, and teenagers to radiofrequency radiation, to avoid an epidemic of early-onset dementia starting in middle age.

- *Death of hippocampal neurons*: The neurophysiological mechanisms of memory are presently ill-understood, but one thing that is known for sure is that a properly functioning hippocampus is essential for the laying down of new memories. Hence the demonstrated loss of hippocampal neurons in teenaged rats exposed to RF [36] reinforces the warning at the end of the preceding subsection.

- *Reproductive damage*: A review of multiple studies on the effects of cell phone radiation on male reproduction [37] reveals that exposure to RF (a) increases oxidative stress and decreases sperm count and motility in rodents; (b) increases oxidative stress, decreases motility, and causes morphometric abnormalities of human spermatozoa in vitro; and (c) does not affect morphology but does cause decreased concentration, motility, and viability of sperm in men using mobile phones, with these abnormalities being directly related to duration of phone use. Fewer studies have been done on female reproduction, but cell phone radiation is reported also to affect the reproduction of female mice by multiple mechanisms [38].

- *Oxidative stress*: Oxidative stress [39] is a condition arising when free radicals (atoms or molecules that have developed unpaired electrons, which make the molecule unstable and highly reactive), outnumber antioxidants (compounds that neutralize free radicals by donating electrons to them). An excess of free radicals, also known as oxidative stress, is implicated in virtually all of the degenerative diseases afflicting humankind: atherosclerosis, heart disease, cancer, inflammatory joint disease, asthma, diabetes, dementia, and degenerative eye disease to name some of them. Oxidative stress also lowers immune function, which impacts the development of infectious diseases. Because low-intensity radiofrequency radiation is now an accepted cause of oxidative stress (for a review of multiple individual studies showing this see [40]), at least some role in the development of all of the above health problems might reasonably be attributed to the radiofrequency radiation in which virtually everyone on Earth is now bathed on a daily basis.

- *DNA damage*: DNA damage caused by non-thermal exposure of cultured cells to RF was one of the earliest reported effects of radiofrequency radiation [41]. For a review of many more recent studies confirming that RF causes DNA damage, see [42].

### 3.3. Misleading Statement Three (p. 53)

"The ICNIRP limits used in the [New Zealand] standard are based on a review of all relevant research on health effects, regardless of the mechanisms that might be involved. ICNIRP and other expert panels that have reviewed the data find that the only effects that show up with any clarity are consistent with the effects of heat stress and occur at exposure levels at which absorption of RF energy in the body (as heat) exceeds the body's ability to dissipate that heat".

In philosophical terms, this is known as an argument from authority. Carl Sagan's view of arguments from authority is: "One of the great commandments of science is "mistrust arguments from authority". . . . Too many such arguments have proved too painfully wrong. Authorities must prove their contentions like everybody else" [43].

The authority in this case is ICNIRP, a small, self-selected, non-governmental organization with known ties to the industry whose expansion it is charged with regulating. The truth of the matter is that at most a few dozen scientists continue to defend the thermal-only paradigm [44]. Five times that number—so far 242 EMF-active scientists from 42 countries—have signed the International EMF Scientist Appeal [45], which calls on WHO, the United Nations, and all member nations to issue health warnings about the risks of EMF exposure and to adopt much stronger exposure guidelines to protect humans and other species from sub-thermal levels of EMF.

### 3.4. Misleading Statement Four (p.39)

"While there is sometimes public concern over the presence of industry representatives on the Committee, in practice they have never attempted to influence the Committee's conclusions on the health effects research and generally see the Committee as a means for them to stay abreast of recent developments. In addition, they are able to bring to the Committee's attention forthcoming developments in their industries that may have policy implications for our Government."

This statement is apparently inserted in an attempt to show that there is no conflict of interest involved in committee deliberations. Unfortunately however, the meeting notes referred to above show that there is no need for industry representatives to influence the committee's conclusions about health research, because the committee is already so compromised that the science is massaged to favor industry interests as a matter of course. The following exerpt from the meeting notes illustrates this:

"Martin Gledhill spoke to his paper on 5G deployment and highlighted the need to ensure that reliable information about the deployment of 5G infrastructure, effects on exposures to

RF fields and health be available ahead of time. Peter Berry [representative of the Electricity Engineers' Association] commented that government and the industry need to work together on this. The Ministry of Health is seen as a credible source of information and should prepare information on health and have this on its website. If the issue develops then ways to communicate more proactively could be investigated."

This underlines the fact that the New Zealand Ministry of Health is, in fact, *not* presently a credible source of information. On the contrary, this government department appears to be firmly and unshakably committed to the ICNIRP thermal-only dogma, exactly because that dogma allows unbridled expansion of the wireless and telecommunications industries.

## 4. Some Hard Numbers: Preliminary Results on Ambient RF Power Densities in Auckland

At present, no raw measurements of ambient RF power density in New Zealand are publicly available. The Telco-funded reports posted on the Ministry of Health website show only percentages of the limits recommended by NZS2772.1:1999, an ICNIRP-inspired Guidelines document which can be purchased from the Standards New Zealand website for $129 + GST. In an attempt to remedy this situation, preliminary measurements were made by the present author in the city center of Auckland, New Zealand in April 2019, using a hand-held Cornet Electrosmog Meter Model EDT 88TPlus.

The results largely fell within the ranges shown for the city centers of Canberra, Sydney, Los Angles, and Addis Ababa in Figure 3 of [46]; i.e. between 2 and 10 mW/m$^2$ (which translates to 0.2–1 µW/cm$^2$). However, two specific hot spots gave cause for concern.

First, the peak reading on the street at the Three Lamps bus stop in Ponsonby Rd at 10:05 on Friday 5 April 2019 was 129 mW/m$^2$. Readings in this location (as with others in the central city) fluctuated rapidly and wildly in time, to the extent that no specific frequency could be recorded. This is perhaps not surprising, considering that the map of cell tower locations available at https://gis.geek.nz/celltowers/ shows three cell towers housing a total of six transmitters, operating at 2100, 700, 850, 1800, 900, and 1800 MHz (a different telco), within 50 m of the bus stop in question, with a further two cell towers housing another four transmitters inside a 100 m radius.

The plethora of different units used by different information sources in this field make comparisons extraordinarily difficult, but a number of online calculators on the web (for example, the one at https://www.compeng.com.au/field-strength-calculator/) reveal that a reading of 129 mW/m$^2$ translates to 12.9 µW/cm$^2$. This is but a tiny fraction of the ICNRIP-based New Zealand exposure guideline of 10 mW/cm$^2$, which translates to 10,000 µW/cm$^2$. However, the 12.9 µW/cm$^2$ recorded at the Three Lamps bus stop is comfortably above the recommended exposure limits of 10 µW/cm$^2$ used by Poland, Slovenia, the Ukraine, Bulgaria, Italy, Switzerland and Brazil, and considerably above the recommended exposure limits of 4.5 µW/cm$^2$ in Canada, 1–10 µW/cm$^2$ in Paris, and 1 µW/cm$^2$ in Lithuania and Salzburg [47]. The 12.9 µW/cm$^2$ recorded at the Three Lamps bus stop is also hugely above the 0.25 µW/cm$^2$ that has been shown to cause oxidative stress and DNA damage in quail eggs [48] and in roughly the same ball-park as the 50–330 µW/cm$^2$ long-term exposure to which has been shown to cause oxidative stress in rat brains [49]. (To the present author's knowledge, no studies on power densities lower than this have been done on whole animals).

A conservative conclusion from these figures suggests that it would be unwise to spend any significant period of time in the vicinity of the Three Lamps bus stop—or indeed in any area of the Auckland central business district, if you consider the quail egg study [48].

The second somewhat disconcerting measurement made in the present, very preliminary study of ambient RF power densities in Auckland was taken immediately outside the door of the microwave oven in the kitchen of the Auckland Council service center in Ostend, Waiheke Island, while the oven was operating. This measurement, which fluctuated between 5.5 and 8.8 µW/cm$^2$, did not quite exceed the safety limits used in most of Eastern Europe, but probably would have been illegal in Canada and Lithuania – and also in Paris and Salzburg. Apparently individual cities can set their own limits on allowable radiation exposure levels. Given that the limits set out in the ICNIRP-inspired New Zealand

standard are not enshrined in New Zealand law, it would also presumably be possible for Auckland City to enforce lower guidelines than those specified in NZS2772.1:1999.

## 5. Which is Worse: Sharp Spikes of RF or Continued Low Level Exposure?

One of the many debatable questions in this area is whether prolonged exposure levels or brief spikes of power are more important in determining the biological effects of RF.

The ICNIRP dogma that tissue heating is the only possible biological effect of RF radiation—which dogma has now been conclusively disproved by several thousand studies, and thus can no longer be considered even a viable hypothesis, let alone a scientific fact—leads to the specification that brief spikes should be ignored and RF measurements should be averaged over 6 min. This approach would indeed be reasonable, if the only effect of interest *were* excessive tissue heating. However, there are now a number of other mechanisms by which RF radiation has been proven to affect biological organisms, at much lower power densities than those needed to heat tissue.

Probably the most important of these mechanisms is overproduction of free radicals, which leads to a cascade of further effects, including DNA damage. In the scientific sense of the term, a free radical is a molecule that has lost one of its electrons and thus become unstable and highly reactive [39]. The mechanism by which RF creates a free radical is likely to involve a single, discrete collision between the molecule in question and a beam of RF. Once created, the free radical can be neutralized again by contact with an antioxidant molecule, which gives back its lost electron; however, if that does not happen immediately, the free radical is capable of producing downstream effects such as damage to DNA, cell membranes, and various other biological entities whose continued function is essential to the organism. This means that RF damage can be mitigated by health-promoting behaviors such as the consumption of antioxidants (vitamin C, fruit and vegetables, dark chocolate). However, it also means that if too many free radicals are created for the natural protective mechanisms of the organism to cope with—or if any given free radical does irreparable damage to some vital biological structure before being neutralized—the effects of RF are cumulative. This analysis leads to the conclusion that both of the factors artificially set in opposition by the question at the start of this section are important. Sharp spikes of RF produce free radicals. Continued sharp spikes of RF produce more free radicals. If enough free radicals are allowed to build up in the body, health impacts become more or less inevitable.

Continued, relentless, sharp spikes of RF are exactly the environmental condition involuntarily imposed on any citizen with the misfortune to live and/or work near one cell tower, let alone five. And oxidative stress of the sort caused by sharp spikes of RF has been shown to contribute to all of the disease states underpinning the increasing number of "epidemics" reported by the media in New Zealand.

## 6. Discussion

A group of investigative journalists known as Investigate Europe allege the existence of an 'ICNIRP cartel': a group of 14 core scientists plus a couple of dozen supporters who act to promote and defend the ICNIRP dogma that the only confirmed harms caused by RF are acute thermal effects [44]. This cartel is alleged to preserve the EMF exposure guidelines favored by industry by conducting biased reviews of the literature, which minimize health risks from exposure to EMF power densities lower than those which cause thermal harm.

The multiple citations to papers and reviews written by ICNIRP members and the many references to ICNIRP beliefs in the text of the report to ministers of the New Zealand government 2018 from the Interagency Committee on the Health Effects of Non-Ionizing Fields reveal that the authors of this report are, for whatever reason, firmly committed to the ICNIRP view. In the service of this view, thousands of papers reporting adverse effects of less-than-thermal RF power densities are simply ignored. When the occasional study is too widely known to be ignored, its significance is minimized and its methodology questioned; but questioned in such a way that no specific, answerable objections

are raised. The conclusion is now inescapable that ICNIRP and its followers are so firmly committed to the thermal-only dogma that no amount of evidence will change their minds.

## 7. Conclusions

- It is time to stop believing ICNIRP spin. Tissue heating is not the only biological effect of radiofrequency radiation. The thermal-only exposure limit is not safe.
- Like tobacco smoke, low intensity radiofrequency radiation has multiple harmful effects on human health. Unlike secondhand smoke, secondhand radiation is fast becoming inescapable. The present situation is thus worse than Big Tobacco redux.
- Elected politicians should stop accepting biased reports from individuals with blatant conflicts of interest and start taking seriously the health and safety of their constituents; or at least of their own children and grandchildren.
- The unchecked expansion of Big Wireless permitted by ICNIRP's thermal-only guidelines is actively harmful to all biological inhabitants of planet Earth. Further expansion to 5G technology will inevitably involve yet more radiation exposure. The fact that this exposure will not breach the ludicrously high ICNIRP-based standard is no defense at all.

**Funding:** No funding has been received from any source for preparation of this paper, or for related work.

**Conflicts of Interest:** The author declares no conflict of interest.

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
