# Peer review of "Conflicts of Interest and Misleading Statements in Official Reports about the Health Consequences of Radiofrequency Radiation and Some New Measurements of Exposure Levels"

_magnetochemistry, doi:10.3390/magnetochemistry5020031_

Round 1

Reviewer 1 Report

This work consists on an interesting point of view related with the exposure protection and public health.

Author should take into account the next recommendations:

1.- In the abstract is better not to use "aka", I think it is better to use "also known as"

2.- Please, be careful with the acronyms, what is NGO? In the abstract and in the text

3.- It should improve the narrative of the article. At the beginning it is a bit confusing, however at the end it imrpves, and it is understood better.

4.-I think that the paper seems more an opinion piece than a scientific article, the article should be rewritten in order to try to show objective data, not opinions.

5.- It should be written again, it is not very clear. The same for the abstract.

6.- To complete the work it is interesting to include differences of the concept of exposure between Western and East European countries. The concept of exposure dose, more popular in Eastern European countries, should be indicated. A distinction must be made between puntual intense exposure and prolonged low intensity exposure.

Author Response

1.- In the abstract is better not to use "aka", I think it is better to use "also known as"

I have replaced aka with 'also known as'.

2.- Please, be careful with the acronyms, what is NGO? In the abstract and in the text

I have replaced NGO with 'non-governmental organization'. 

3.- It should improve the narrative of the article. At the beginning it is a bit confusing, however at the end it imrpves, and it is understood better.

This comment itself is more than a bit confusing.  Referee 1, who (like the writers of the Report to Ministers that is critiqued in the present paper) chooses to remain anonymous, first checks the box saying he or she does not feel qualified to comment on the English; then makes the major comment of giving the paper one star for expression; then says (in rather artfully fractured grammar, but using the Western-political buzz-word 'narrative') that he can't understand the beginning; but does not say why.  Which bits of the beginning are incomprehensible?  I am left to guess.

My first guess is that perhaps this referee does not understand the term "conflict of interest".  That actually seems fairly likely, so I have added a paragraph at the start of the relevant section of the manuscript defining the concept. I think this addition improves the paper.

[That being said, it remains a mystery to me why scientific journals, which these days quite properly do ask their authors to declare any conflict of interest, still fail to ask their reviewers to make any similar declaration -- and indeed still allow those reviewers to remain anonymous.  What is the justification for demanding that an author who has put significant work into producing a properly documented and researched paper should respond to non-specific, undocumented negative comments from a reviewer who is not even willing to sign his name to his remarks?].

To return guessing the meaning of the vague comment at issue here, perhaps it is simply my vocabulary that defeats the referee.  The word endemic may be unfamiliar?  I have replaced "conflict of interest has been endemic at all levels..." with "conflict of interest has been a constant feature at all levels ..." 

In the absence of further guidance, I am at a loss to know quite what the reviewer's problem is and conclude that taking the arithmetic mean of his one star and referee 2's five stars gives three stars, which indicates that the language in the paper is acceptable. 

4.-I think that the paper seems more an opinion piece than a scientific article, the article should be rewritten in order to try to show objective data, not opinions.

My first response on reading this was to point out that this remark itself (like the reviewer's previous remark) is clearly an opinion rather than a scientific comment and to suggest that perhaps it, rather than the article, is what "should be rewritten in order to try to show objective data".  But that seemed a bit childish. 

My second impulse was to take the line of least resistance and fall into the reviewer's trap by reclassifying the paper as an Opinion piece.  But just in time, I remembered that one of the top five strategies in the ICNIRP playbook is to dismiss perfectly well-documented scientific papers as merely "opinion" and thereby justify ignoring them. 

In the end I decided to answer Point 4 by acceding to the referee's demand that I rewrite the paper showing objective data.  This I have done, by adding a new section in which I report my own pilot measurements of the radiofrequency power density at two locations in Auckland City. The resulting objective data on local exposures are then compared with (a) the ICNIRP-inspired official safety guidelines defended by the specific governmental report critiqued earlier in the present paper (b) the official safety guidelines used in Eastern Europe, Italy, Switzerland, Brazil, Canada and various other jurisdictions and (c) the power densities shown in specific scientific papers to cause adverse health effects in the laboratory. I believe the result is actually a valuable addition to the debate and thank the referee for taking the typically bureaucratic action of delaying his review until just before the stated deadline, thus unintentionally allowing me time to purchase the necessary RF meters.

5.- It should be written again, it is not very clear. The same for the abstract.

I hope the extensive changes I have made as above have now made the paper very clear.  As part of these changes, the abstract has been completely rewritten and the introduction partially rewritten to take account of the new sections of the manuscript. I think these changes have improved the manuscript significantly and  hope the journal editor can now fulfill their responsibility to make an informed judgement and will decide to publish the paper.

6.- To complete the work it is interesting to include differences of the concept of exposure between Western and East European countries. The concept of exposure dose, more popular in Eastern European countries, should be indicated. A distinction must be made between puntual intense exposure and prolonged low intensity exposure.

Yet again, the basis of these completely undocumented (and arguably quite muddled) remarks is unclear. Eastern European safety limits are expressed in exactly the same terms as Western limits -- as power densities. Power density is a measure of power (in watts or milliwatts) per unit area (in square centimetres or square metres). The only difference between Eastern European and Western countries in this regard is that Eastern power density limits are science based, while most Western limits are dogma based. As a consequence, the Eastern limits are set at very much lower levels than the Western limits.

The question of whether it is more appropriate to average dosage over time (as ICNIRP dogma demands) or to pay more attention to maximum spikes of energy (as anyone who has killed an animal with a single bullet might be inclined to do) are discussed in another new section of the paper, entitled Which is worse: sharp spikes of RF or continued low level exposure? I thank the referee for the opportunity to address these important issues.

Reviewer 2 Report

Great paper and true fact.

Author Response

Great paper and true fact.

Thanks.

Round 2

Reviewer 1 Report

I recommend the paper for publication

Magnetochemistry EISSN 2312-7481 Published by MDPI AG, Basel, Switzerland RSS E-Mail Table of Contents Alert
Back to Top